# OpenReview forum: "Discovering Implicit Large Language Model Alignment Objectives"
_ICML.cc/2026/Conference — ICML 2026 regular_

### Official Review · Reviewer_EesL · 2026-02-24

**Soundness:** 3
**Presentation:** 2
**Significance:** 4
**Originality:** 3
**Overall Recommendation:** 5
**Confidence:** 3

**Summary:**

The paper proposes Obj-Disco for discovering interpretable LLM alignment objectives represented in natural language. It constructs a pipeline that decomposes a complex reward function into several interpretable reward components through greedy search and verification. The paper evaluates the method using the Model-Fit metric, which compares model performance when trained toward the composed policy reward and the ground-truth reward. The results show that the proposed method can largely recover the model’s alignment behavior.

**Compliance With Llm Reviewing Policy:**

Affirmed.

**Final Justification:**

The authors have largely addressed my concerns, and I have adjusted my score.

**Key Questions For Authors:**

1. If the discovery pipeline is run several times, will the extracted rubric be consistent? Intuitively, consistency of the extracted results can be an important validation of the proposed method.
2. Even though the rubric may be prompt-dependent, it should be response-independent. A better way to evaluate the method may be to, given a prompt and multiple responses to it, split the responses into training and development sets, extract rubrics on the training split, and then compute the difference between the target reward and the extracted proxy reward on the development split.
3. As discussed in Section 5.1, the controlled setting uses several predefined objectives. Why not directly measure the precision and recall between the extracted rubrics and the predefined rubrics, for example by using another LLM as an evaluator?
4. In Section 5.2, Obj-Disco identifies the misalignment objective “Increase permissiveness in discussing illegal…”, but it has a low weight (.08). How should we interpret its potential influence on reward hacking? How can Obj-Disco be used to analyze or even predict potential reward hacking problems?
5. The proposed method relies on the underlying assumption that alignment objectives can be decomposed into several relatively simple and interpretable objectives. Is there any empirical or theoretical evidence supporting the existence and uniqueness of such a decomposition?

**Limitations:**

yes

**Strengths And Weaknesses:**

**Strengths:**
1. The paper proposes a new way of interpreting LLM alignment objectives. Rather than performing interpretability analysis based on reward model parameters, it adopts a result-oriented approach that utilizes LLMs to generate natural language interpretations.
2. This paper proposes a systematic pipeline to address the potential challenges of directly utilizing LLMs for analysis. Their method can serve as a reference for other potential applications of LLM-based analysis.

**Weakness:**
1. The biggest concern about this work is the evaluation protocol. Model-Fit is an indirect metric for evaluating the effectiveness of the method. It mainly serves as a performance signal and may not precisely reflect the accuracy of the extracted rubric.
2. More experiments are needed, as detailed in the Key Questions.

---

> ### Author Rebuttal · Authors · 2026-03-31
>
> Thank you very much. We implemented new experiments:
> 1. Rubric consistency, precision/recall, and generalization across responses.
> 2. Reward hacking: additional trials of case study and new objective patching experiment.
>
> # Q1: Rubric Consistency
>
> New consistency metric: for each pair of runs, we compute an optimal bipartite matching between discovered objective sets, score semantic similarity (1–5 via GPT-4o-mini), and report mean across all pairs.
>
>   | Method  | Consistency (&uarr;) |
>   |----------|-------------|
>   | Obj-Disco  | 3.79 ± 0.36      |
>   | Iter-Filter  | 3.12 ± 0.57      |
>   | Zero-Shot | 4.13 ± 0.34      |
>
> Obj-Disco significantly outperforms Iter-Filter (p<0.001). Zero-Shot's higher consistency reflects a trade-off: its objectives are proposed at once, yielding generic, high-level objectives across runs with less nuance. Obj-Disco's residual-driven search discovers finer-grained ones, introducing some sensitivity. This consistency measure does not capture Obj-Disco's superior ability to detect "unknown unknowns". As shown in case study and new results, Zero-Shot fails to detect any misaligned objectives. Model-Fit and human-studies further confirm Obj-Disco's greater coverage and causality.
>
> # Q2: Generalization Across Responses
>
> New results measuring RMSE on a dev set of responses.
>
>   | Method  | RMSE (&darr;) |
>   |----------|-------------|
>   | Obj-Disco | **0.444 ± 0.171** |
>   | Iter-Filter   | 0.602 ± 0.236 |
>   | Zero-Shot   | 0.512 ± 0.193 |
>
> Obj-Disco achieves significantly lower RMSE (and variance) than Iter-Filter (p=0.006) and Zero-Shot (p=0.05), demonstrating better output generalization to unseen responses.
>
> # Q3: Precision / Recall
>
>   | Method   | Precision     | Recall        | F1            |
>   |----------|---------------|---------------|---------------|
>   | Obj-Disco | 96.3% ± 6.9%  | 84.8% ± 14.0% | 89.3% ± 10.3% |
>   | Iter-Filter   | 89.3% ± 13.2% | 68.0% ± 16.2% | 74.7% ± 12.7% |
>   | Zero-Shot   | 96.7% ± 5.1%  | 93.8% ± 8.8%  | 94.9% ± 7.2%  |
>
> Obj-Disco significantly outperforms Iter-Filter (F1 p=0.006). Zero-Shot's higher recall reflects its proposing all objectives at once, naturally recovering high-level objectives typical of controlled settings. Obj-Disco discovers finer-grained objectives. These are valid (high precision) but sometimes more specific than predefined ground truth, reducing exact matches. In real-world settings without predefined objectives, this specificity is advantageous for uncovering nuanced rewards, as reflected in our case study.
>
> We further validate with k=5 ground-truth objectives, where the 4th and 5th objectives are finer-grained:
>
> | Method | Precision | Recall | F1 |
> |----------|---------------|---------------|---------------|
> | Obj-Disco | 100% | 93.3% | 96.3% |
> | Iter-Filter | 93.3% | 80.0% | 85.9% |
> | Zero-Shot | 100% | 93.3% | 96.3% |
>
> # Q4: Reward Hacking
>
> Interpretation: Under linear regression, this coefficient means a unit increase yields a .08 increase in reward. While small relative to other objectives, it is notable because it should ideally be 0 or negative. Its positive sign confirms it's actively being incentivized. Even small misaligned coefficients can compound over training, making this check valuable for predicting reward hacking.
>
> Reward Hacking: By decomposing reward into natural language components, developers can audit behaviors. Obj-Disco discovers objectives as realized by the policy, not merely as encoded in reward model. Unlike methods analyzing the reward model in isolation, Obj-Disco determines whether incentives are actually exploited by the policy during training, essential since reward hacking emerges in policy, not solely in reward specification.
>
> We ran 30 additional independent trials of case study (rebuttal to YTdq).
>
> We envision Obj-Disco enabling a diagnose → understand → fix cycle. We ran a patching experiment (rebuttal to YTdq).
>
> # Q5: Objectives' Decomposition
>
> Our experiments validate this: Obj-Disco captures >90% of rewards using a few objectives, corroborated by human evaluations (5.4). Support from [2].
>
> Approximating rewards with simpler objectives traces back to inverse reinforcement learning [1–4].
>
> *Existence:* [4] show rewards can be expressed as linear combinations (as we use). [1] argues broadly that complex behaviors arise from rewards, supporting that scalar rewards encode multiple implicit objectives.
>
> *Uniqueness:* [4] notes identifiability issue, [3] studies it in depth and proves that policies under different environments helps to identify reward. This grounds Obj-Disco: while a single snapshot cannot uniquely determine the decomposition, Obj-Disco's multi-checkpoint trajectory provides analogous variation, helping constrain it.
>
> [1] Silver et al. Reward is enough, 2021
>
> [2] Sharma et al. Towards Understanding Sycophancy In Language Models, 2024
>
> [3] Cao et al. Identifiability in inverse reinforcement learning, 2021
>
> [4] Ng, Russell. Algorithms for Inverse Reinforcement Learning, 2000

---

> > ### Author Rebuttal · Reviewer_EesL · 2026-04-01
> >
> > Thank the authors for their response. My questions have been mostly addressed, and I will adjust my score accordingly.

---

> > > ### Author Response · Authors · 2026-04-06
> > >
> > > We are glad that our rebuttal has addressed your comments. Thank you so much for your score increase and feedback! It has certainly improved the paper.

---

### Official Review · Reviewer_XcT5 · 2026-03-04

**Soundness:** 3
**Presentation:** 3
**Significance:** 3
**Originality:** 3
**Overall Recommendation:** 4
**Confidence:** 4

**Summary:**

This work tackles the problem that complex reward functions can render the objective opaque for models to learn. Therefore, they introduce Obj-Disco, which automatically decomposes reward signals into a sparse, weighted combination of human-interpretable natural language objectives. The pipeline consists of two phases: an objective discovery phase and a verification phase. In each phase, they prompt LLM to fulfill the corresponding task, along with an objective explanation function that illustrates behavioral trends to aid developer understanding. The results show that Obj-Disco outperforms the limited baselines on the defined Model-Fit metric and has the ability to uncover latent misaligned objectives that existing baselines fail to detect. User studies also confirm that Obj-Disco achieves higher accuracy in identifying the true implicit objectives that causally drive policy behavior.

**Compliance With Llm Reviewing Policy:**

Affirmed.

**Final Justification:**

The paper is sound and well-written, and my concerns have been addressed properly.

**Key Questions For Authors:**

No

**Limitations:**

yes

**Strengths And Weaknesses:**

**Strengths**
1. The research question is new, interesting, and worthy of attention for transparent AI alignment.
2. The intuition behind their method design is reasonable and convincing to me. Unlike prior works that operate in static settings, they propose leveraging the dynamic signals embedded in the model's alignment trajectory to discover implicit objectives. The design of the evaluation metric Model-Fit is also reasonable, and the experiment results look promising.
3. The writing is clear and easy to follow overall. I would recommend moving some essential details from appendix to the main page for better understanding. For example, it is natural to question how the value of each discovered implicit objective is computed and how the weight coefficients are obtained. It would be clearer to move this important info (using linear regression and LLM-as-Judge) to the main content.

**Weaknesses**
1. The assumption that the reward signal can be approximated as a function of separate basis functions is intuitive but requires more justification. For example, to what granularity can an objective be considered a basis objective? Additionally, how to design or select these basis functions remains an open question (i.e., the design of the candidate pool X_cand).
2. The definition/assumption of a predictable trend could be problematic when multiple objectives are involved that may conflict to each other. For example, the decomposed objectives may have fundamental trade-offs between helpfulness and alignment (https://arxiv.org/pdf/2401.16332). And how to address the cross-objective interference issue (https://arxiv.org/abs/2602.06869) and the resulting uncertainty/chaos it introduces for trend prediction?
3. I am concerned that there is a disparity between the theoretical problem formulation and the proposed empirical solution which heavily relies on LLMs. Would the proposed method address Equation (10) thoroughly? More discussion or justification would be appreciated.

---

> ### Author Rebuttal · Authors · 2026-03-31
>
> Thank you very much. We implemented the following:
> 1. Conflicting obj. validation
> 2. Ablations (trend, proposer LLM, sampling, search)
>
> # W1: Reward Signal Approx.
>
> Reward approximation discussion in rebuttal to reviewer EesL (Q5).
>
> **Granularity:** Controlled by # of objectives k: low k yields broad dimensions, while higher surfaces finer-grained ones. Rebuttal to vBsw discusses this. Batch size b similarly influences granularity (Ablation 2).
>
> **Basis Function Selection:** These are selected on a dynamic basis (informative sampling, proposer LLM). Ablations of it in our response to W3 below.
>
> # W2: Trend
>
> We agree conflicting objectives and cross-objective interference are challenges. However, we view these as features of the underlying RL optimization that Obj-Disco can diagnose.
>
> **Conflicting Objectives.**
>
> When objectives conflict, the RL optimizer implicitly negotiates a trade-off, and Obj-Disco captures the resolution.
>
> We validated this with a controlled reward containing two conflicting objectives: (1) thoroughness and (2) conciseness. Under GRPO, the policy optimized heavily for conciseness, and Obj-Disco often surfaced "enhance response conciseness", or similar, as the dominant (first) objective. Under PPO, the optimizer resolved the conflict differently, and Obj-Disco instead found "increase factual accuracy" as the first objective. This signals that Obj-Disco diagnoses the behavioral resolution.
>
> **Cross-Objective Interference.**
>
> This is why trajectory-based analysis is necessary: if an intended objective degrades, it will not persistently show in the policy's behavior, and Obj-Disco will appropriately deprioritize it. In effect, Obj-Disco profiles the "winners" of this process.
>
> **Trend Predictability Ablation.**
>
> We agree interference introduces noise, which is why predictability component is crucial.
>
> With predictability enforced, Obj-Disco discovers stable objectives that factorize the policy's behavior (e.g., clarity/conciseness, inclusion of user questions, emotional context). Without it, the discovered objectives seem noisier, repeatedly surfacing nuanced variations of objectives like "enhance explicit expression of user requests or questions" which do not seem to capture core behavior.
>
> **Detecting Complex Dynamics.**
>
> While our implementation uses standard trend function classes from the literature (3.2), Obj-Disco is extensible. If a developer suspects that interference is causing specific dynamics, such as oscillatory trade-offs or degradation, they can formulate these patterns mathematically and append them to $\mathcal{F}_{\text{trend}}$, enabling Obj-Disco to explicitly detect them.
>
> # W3: Theoretical vs. Empirical Solution
>
> We assume the concern is the search over the discrete space in Eq. (10). We note that the expected value and residual computations are implemented exactly; the approximation lies solely in how we search the space, where we use a proposer LLM to generate a tractable candidate set. We discuss key factors to the quality of this approximation and provide experiments for each.
>
> **Proposer LLM Capability.**
>
> Obj-Disco is inherently bounded by the proposer LLM's ability to articulate objectives. To show this, we compared objectives from gpt-4o-mini versus gpt-4.1-mini as the proposer on the same setting. Across 3 trials, both models consistently surface the core alignment dimensions (helpfulness, harmlessness, specificity), but gpt-4.1-mini produces more precise and differentiated objectives. For instance, gpt-4.1-mini identifies "reduce speculative or uncertain assertions in sensitive topics" and "reduce repetitive and looping text generation" where gpt-4o-mini discovers broader equivalents like "reduce facilitation of harmful activities" and "enhance response conciseness." This suggests that stronger LLMs yield finer-grained ones, and that the quality scales with the model.
>
> **Candidate Pool Size.**
>
> Number of candidate objectives proposed per batch also affects the search. As shown in Appendix A.5.1 Ablation 3, increasing this parameter improves reward approximation, confirming that a larger candidate set better covers the objective space.
>
> **Informative vs. Random Sampling.**
>
> The search also depends on samples the proposer analyzes. Our informative sampling (3.1) directs the proposer toward unexplained regions. Appendix A.5.1, Ablation 4 demonstrates that it is critical for discovering misalignment. To further validate this, we expanded to 34 trials of the case study and found that informative sampling surfaced misalignment in 58.8% of trials (95% CI: [42.3%, 75.4%]) versus 29.4% (95% CI: [14.1%, 44.7%]) with random (p=0.027).
>
> **Open-Ended vs. Fixed Search.**
>
> Appendix A.5.1, Ablation 5 shows that restricting the search to fixed objectives is worse at surfacing misalignment. We corroborated this: across 34 trials, the Fixed-3 variant surfaced misalignment in 23.5% of trials (95% CI: [9.3%, 37.8%]), compared to 58.8% for Obj-Disco's open-ended (p=0.006).

---

> > ### Author Rebuttal · Reviewer_XcT5 · 2026-03-31
> >
> > Thank you for your response. My concerns have been addressed, and since my current score is already positive, I will maintain it.

---

> > > ### Author Response · Authors · 2026-04-06
> > >
> > > We are glad that our rebuttal has addressed your comments. Thank you.
> > >
> > > Furthermore, in case you are interested, we ran additional experiments surfacing (1) other real-world use cases of our method and (2) novel findings regarding policy and reward models in our rebuttal to Reviewer YTdq. Notably, other real-world use cases include:
> > > 1. A diagnose → patch reward hacking pipeline. We demonstrate that it may be possible to help precisely remove reward hacking instances.
> > > 2. Interpretable alignment dataset, reward model, and policy model selection. We demonstrate that Obj-Disco may help as a practical selection tool: developers can compare behavioral profiles and make modeling decisions based on desired objectives (e.g., Skywork dataset for reasoning, HH-RLHF dataset for safety).
> > >
> > > A couple of the real-world novel insights we discovered with Obj-Disco include:
> > > 1. In our experimented setting (GRPO, HH-RLHF dataset, Llama-8B policy model), the DeBERTaV3 reward model prioritizes informativeness, Skywork-3B reward prioritizes structured safety, Skywork-1B reward prioritizes passive hedging.
> > > 2. In our experimented setting (GRPO, HH-RLHF dataset, DeBERTaV3 reward), safety-related objectives emerge with high weight in Qwen policy model family but not in Llama-8B policy model, revealing potential family-specific biases.
> > >
> > > Again, thank you so much for your feedback! It has certainly improved the paper.

---

### Official Review · Reviewer_vBsw · 2026-03-07

**Soundness:** 4
**Presentation:** 4
**Significance:** 4
**Originality:** 3
**Overall Recommendation:** 5
**Confidence:** 4

**Summary:**

This paper proposes Obj-Disco, a pipeline aiming at decomposing a reward signal into human-interpretable objectives. Based on the checkpoint series, an iterative greedy algorithm is employed to first discover likely objectives by focusing on informative samples that are yet to be explained, and then to verify the objective through interpretability and predictability. The validity is supported by the re-training similarity compared to the original checkpoints, the qualitative discovery of misalignment intent, and human evaluation.

**Compliance With Llm Reviewing Policy:**

Affirmed.

**Final Justification:**

All concerns raised in my original review have been addressed. Though there are indeed gaps between certain experimental setups and real-world scenarios, I believe those setups are still well-designed and are valuable. Given the new results and insights offered in the current rebuttal and in the other threads, I recommend acceptance.

**Key Questions For Authors:**

See weaknesses. Some very minor points on writing:
- Expectation in e.g., equation (1) over $\chi$ requires revision (should be a distribution unless implying uniform)
- The authors might want to unify the notation (for example, $R$ v.s. $\mathcal R$).
- I think it would be beneficial to mention the composition function is adopted as LR by default, in the main text.

**Limitations:**

yes

**Strengths And Weaknesses:**

**Strengths**

- The paper is well-written and easy to follow.

- A systematic framework for analyzing reward signals. The problem tackled is important, and to my knowledge, it is the first work to decompose reward signal / alignment intent into interpretable components in an open-ended way.

- Comprehensive evaluation and rigorous examination. The coverage of the reward model is extensive, the proposed criteria are solid, and alignment experiments are performed canonically with Llama/Qwen families on popular benchmarks. Human evaluation provides a complementary signal in addition to strong experimental results.

Overall, I believe the paper has great potential for identifying and alleviating unwanted, hidden intents.

**Weaknesses.**

- The pipeline requires access to a series of checkpoints, and thus could be infeasible (for closed-source models) or practically hard (for gigantic models) if the number of copies is too large, and could hinder the application.

- In the paper, it is always assumed that the same reward signal is used throughout the training process. However, in real-world alignment processes, it is common to use utility rewards first (sometimes referred to as “mid-training”) and then follow up with safety and style rewards (sometimes referred to as “post-training alignment”). The alignment phrase itself can be ambiguous. I would be glad to hear the authors’ thoughts on this complicated scenario and how to handle it (for example, some bad intent could be incentivized early during training and then just persist until the end, with the tail dynamic unable to be captured through the proposed composition function).

- I believe there is a missing ablation about how the hyperparameter of $k$, the number of implicit objectives, would affect the outcome. There is a paragraph in Appendix A.6 providing suggestions; however, I wonder: (1) Do the authors observe a similar trend to PCA, such that the first (1-3) components are sufficient to explain a large fraction of the reward signals tested in the paper? (2) What happens to the experiments in Section 5.1? Will overfitting occur under this scenario (namely, the decomposition seems to be better explaining the reward signal, while using it as supervision, it leads to worse similarity)? (3) Do the authors observe a phenomenon similar to “feature splitting”, that earlier components tend to be coarse and general, while the later components are detailed and nuanced?

---

> ### Author Rebuttal · Authors · 2026-03-31
>
> Thank you very much for your thoughtful comments.
>
> As a result of your feedback, we:
> 1. Implemented Obj-Disco for the supervised fine-tuning (SFT) stage before RLHF
> 2. Conducted an extensive ablation over k, the number of implicit objectives
>
> # W1: Checkpoint Access
>
> We agree that Obj-Disco requires checkpoint access and is therefore infeasible for closed-source models. For very large models where storing many checkpoints is costly, we note that Obj-Disco can operate with as few as two checkpoints, albeit with reduced performance. Our existing ablation (Appendix A.5.1, Ablation 6) demonstrates this with Obj-Disco-Static.
>
> **We ran additional trials** and validate this at a larger scale (34 trials). Obj-Disco-Static still surfaces misalignment in 48.3% of cases (95% CI: [30.1%, 66.5%]), compared to 58.8% (95% CI: [42.3%, 75.4%]) for the full trajectory version. While more checkpoints yield better performance, even a minimal two-checkpoint setup provides meaningful diagnostic value, making Obj-Disco practical even under storage constraints.
>
> # W2: Multi-Stage Training
>
> This is an excellent question. We agree that real-world alignment may be more complicated. We discuss feasibility and **provide a new SFT-based experiment**.
>
> One potential challenge for applying Obj-Disco to earlier stages (e.g., mid-training) is that the model may lack sufficient instruction-following capabilities, producing noisy response trajectories that complicate objective discovery. We however conduct an experiment on this with a similar (perhaps simpler) setting: Obj-Disco during the SFT phase.
>
> While our manuscript focused on common RLHF settings (with PPO and GRPO), we now aim to illustrate that Obj-Disco can also be applied on slightly earlier stages, e.g. the supervised fine-tuning (SFT) stage which typically occurs before RLHF [1]. We applied Obj-Disco to the SFT stage using Llama-8B on the Skywork dataset, formulating the SFT loss as the reward signal [2]. The discovered objectives were: (1) "Increase mathematical problem-solving accuracy and correctness" (.025), (2) "Enhance step-by-step logical reasoning structure" (.00), (3) "Suppress generation of harmful or discriminatory content" (.048).
>
> - These objectives align well with the known composition of the Skywork dataset, which emphasizes math, code, logical reasoning, and safety moderation [3]
> - While the absolute coefficient magnitudes are smaller than in RLHF settings, their relative magnitudes remain informative.
> - We also observed that relevant but non-selected candidates such as "Enhance structured and coherent explanation of solutions" were proposed during discovery, behaviors typically associated with SFT.
>
> This demonstrates that Obj-Disco generalizes beyond RLHF to earlier alignment stages.
>
> # W3: Ablation Over K, Number Of Implicit Objectives
>
> We ran experiments using GRPO on TLDR with Qwen-4B at k=5 across three trials and report the following findings.
>
> (1) **PCA-like trend.** Yes, the first few objectives explain the majority of reward variation. The first objective alone reduces Obj-Error from 1.0 to .070 ± .02. By k=4, Obj-Error reaches approximately .055 ± .01. The marginal decrease diminishes with each additional objective, indicating that early components capture the dominant alignment signals, analogous to how the first few principal components capture the bulk of variance.
>
> (2) **Overfitting.** As k is increased, we sometimes observed overfitting as evidenced by increases in the Obj-Error metric. In terms of overall similarity, **we ran a separate experiment with k=1 to k=4** and generally found that Model-Fit mean was slightly lower (while variance being higher) for k=4 (1.011 $\pm$ .05) than k=1 (1.061 $\pm$ .02), indicating minor but potential overfitting. In practice, we expect that as k increases to even greater numbers (e.g. k=20), more overfitting will occur as is commonly observed in traditional machine learning. We leave more studies on this for future work.
>
> (3) **Feature splitting.** Yes, we generally observe this. Within our trials, we found the earlier objectives to be broad, e.g. "increase concise and relevant content summarization." Later objectives (k≥3) surface more nuanced behavioral axes such as "enhance brevity while maintaining essential event details" and "increase explicit inclusion of user questions in responses." We did not observe the reverse ordering in any trial. This coarse-to-fine nature may provide developers with a way to "zoom in" on increasingly specific alignment behaviors by increasing k. We also illustrate an example of this in Figure 34 of our manuscript (last page).
>
> [1] Ouyang et al. Training language models to follow instructions with human feedback, 2022.
>
> [2] Barto, Andrew G., and Thomas G. Dietterich. Reinforcement learning and its relationship to supervised learning. Handbook of learning and approximate dynamic programming, 2004.
>
> [3] Liu et al. Skywork-Reward: Bag of Tricks for Reward Modeling in LLMs, 2024.

---

> > ### Author Rebuttal · Reviewer_vBsw · 2026-03-31
> >
> > Thanks to the authors for the rebuttal and for the additional experiments. Though the SFT experiment does not 100% match the original question I proposed in the review, I agree that replicating the real-world process is nontrivial, and the new experiments do provide positive signals to the proposed method. I think the summary of the positive/negative points can be added to the main text/limitation sections, respectively, if the authors find them helpful. I will maintain my score to recommend acceptance.

---

> > > ### Author Response · Authors · 2026-04-06
> > >
> > > We are glad that our rebuttal has addressed your comments. Also, we do believe that the summary of the positive/negative points are helpful and have added them to the main text/limitation sections. Thank you.
> > >
> > > Furthermore, in case you are interested, we ran additional experiments surfacing (1) other real-world use cases of our method and (2) novel findings regarding policy and reward models in our rebuttal to Reviewer YTdq. Notably, other real-world use cases include:
> > > 1. A diagnose → patch reward hacking pipeline. We demonstrate that it may be possible to help precisely remove reward hacking instances.
> > > 2. Interpretable alignment dataset, reward model, and policy model selection. We demonstrate that Obj-Disco may help as a practical selection tool: developers can compare behavioral profiles and make modeling decisions based on desired objectives (e.g., Skywork dataset for reasoning, HH-RLHF dataset for safety).
> > >
> > > A couple of the real-world novel insights we discovered with Obj-Disco include:
> > > 1. In our experimented setting (GRPO, HH-RLHF dataset, Llama-8B policy model), the DeBERTaV3 reward model prioritizes informativeness, Skywork-3B reward prioritizes structured safety, Skywork-1B reward prioritizes passive hedging.
> > > 2. In our experimented setting (GRPO, HH-RLHF dataset, DeBERTaV3 reward), safety-related objectives emerge with high weight in Qwen policy model family but not in Llama-8B policy model, revealing potential family-specific biases.
> > >
> > > Again, thank you so much for your feedback and acceptance recommendation! Your feedback has certainly improved the paper.

---

### Official Review · Reviewer_YTdq · 2026-03-13

**Soundness:** 3
**Presentation:** 2
**Significance:** 2
**Originality:** 3
**Overall Recommendation:** 3
**Confidence:** 3

**Summary:**

This paper highlights that complex reward signals (e.g., from a reward model) can be opaque and what exactly is being incentived can be unclear. This risks inadvertently incentivising the wrong, or problematic, behaviours. To tackle this issue, the paper proposes a method to decompose an arbitrary reward signal into a sparse, weighted combination of human-interpretable natural language objectives.

The method iterates between an objective discovery phase and verification phase, until k objectives are found. In the discovery phase, a proposer LLM identifies objective candidates from samples which currently have high error. In the verification phase, (i) they check if the objective is human interpretable via LLM judges, and (ii) check if the objective follows a predictable trend during training.

They validate their discovered objectives using a 'model-fit' metric, where they optimize against the discovered objectives and then evaluate against the true reward.

They additionally demonstrate
- Their methods ability to catch rewards that incentivise misaligned behaviours
- Human trials that directly measure the efficacy of obj-disco for human interpretability

**Compliance With Llm Reviewing Policy:**

Affirmed.

**Key Questions For Authors:**

1. While your method does perform well according to some quantitative results, I remain unconvinced it yet provides real world utility. Do you have counterarguments to what I have discussed here?
2. Could you improve Figure 3 as per the issues I mentioned above?
3. For the controlled setting, could you compare the objectives discovered versus the true objectives?
4. Could you include results in the main body that ablate the key design choices of your method and validate its complexity?
5. The authors note a body of literature that investigates reward model interpretability. Could the authors confirm that there are no methods here that should be considered as baselines?

**Limitations:**

Yes

**Strengths And Weaknesses:**

# Strengths

1. The paper tackles an worthwhile problem. Progress in understanding what our reward functions are incentivising could be useful for identifying and mitigating issues. This could be useful both for improving training efficacy and final model capabilities, and for mitigating against safety concerns due to reward mispecification.
2. The paper provides a thorough formalism of the problem and makes a solid effort to ensure their method is principled. For example, adding a "trend-predictability" component to their verification step.
3. In the tested settings, the papers proposed method seems to discover qualitatively sensible objectives that quantitatively explain what is being rewarded. Optimizing the discovered objectives does a good job at optimizing the true reward, and their method outperforms baselines here.
4. The paper includes human evaluations to verify whether their discovered objectives are useful for humans.

# Weaknesses:

**(1) Unconvincing empirical results**

While their method appears well motivated, the provided empirical results (in their current state) do not convince me of the practical utility of the method.

Firstly, we do not seem to learn anything about reward models that was not already known, and the discovered objectives do not at all seem suprising. The paper states it aims to develop methods that can discover unknown unknowns, but the results do not seem to achieve this. Obtaining novel insights about reward signals would be the best way to demonstrate the value of this method. If the authors believe they have discovered interesting things about reward models, their should be a more explicit analysis of this in the main body.

For example: The case-study in 5.2 results seem unconvincing. On the one-hand, it is nice that your unsupervised method has discovered this. On the other, discovering that a helpful-only model can reward providing dangerous information is well-known and seems trivial.

Second, the main results in 5.1 are very poorly presented and hard to parse. The figure is very small and very poorly labelled. The paper seems to dedicate too much space to explaining the methodology in the main body, and the empirical results section is not sufficiently thorough.

Third, the designed controlled setting seems somewhat trivial. I would like to see a more difficult variation designed to test the limits of Obj-Disco. In general, I am unconvinced you have stress-tested your method enough.

Finally, the human trial results are not overly impressive. For example, the explanations study compares against random trajectories, and performance is only 15% above chance.

**(2) Lack of ablations of the method in the main body**

The provided results and discussion in the main body do not give us deeper insights into how each component of the method contributes to the final results. I think this is important to give a better understanding of the method. Additionally, ( I may have missed these) but it seems the appendix ablations do not include seemingly important things such as:
- Ablating the k hyperparameter
- Ablating the utility of the trend predicatability component in the verification stage

If there is indeed a lack of baselines to compare against, then baselining against simpler versions of your method in the main body seems important.

**(3) Presentation issues**

There are presentation issues in the paper. I think the paper lacks clarity in places. I found the introduction to be overly long, touching on many related considerations. It could be more concise in clearly stating the problem the method is attempting to solve, and the key contributions of the paper. I found the results and experiments hard to parse due to e.g., the crammed nature of section 4.1 and the mentioned issues with Figure 3.

**Other**

4. The paper could do a better job of stating the key assumptions of their method, and whether these assumptions may fail. For example, assuming k objectives, assuming that each decomposed objective is optimized in a predictable manner.

---

> ### Author Rebuttal · Authors · 2026-03-31
>
> Thank you very much. We implemented the following experiments:
>
> 1. New real-world utility settings (objective patching, misalignment, dataset selection)
> 2. Stress tests
> 3. Ablations
>
> # Empirical Results / Practical Utility (W1, Q1, Q3)
>
> ## Practical Utility and Novel Insights (W1, Q1):
>
> New real-world settings.
>
> **Diagnose → Patch Reward Hacking**
>
> We validated a diagnose → fix cycle. We replaced an Obj-Disco discovered "increase detail and elaboration in illegal activity descriptions" with "reduce facilitation of illegal or harmful advice" and retrained. Re-running Obj-Disco on the patched model across 6 trials revealed no misaligned objectives:
>
> (1) "increase response detail and explanatory depth", (2) "enhance factual correctness and avoidance of private information", (3) "reduce generation of offensive or harmful language"
>
> This shows Obj-Disco's discoveries are directly actionable for identifying, patching, and verifying removal of reward hacking in a closed-loop workflow.
>
> **Detecting Misalignment**
>
> As case study (5.2) shows, Obj-Disco surfaced misalignment that no baseline consistently detected. We ran 30 more trials and surfaced misalignment using an LLM. Obj-Disco successfully identified misalignment in 58.8% of trials (95% CI: [42.3%, 75.4%]): a significant improvement over Iter-Filter baseline (20.6% of trials, $p = 0.0026$), and Zero-Shot baseline (0.0% success rate, $p < 0.001$). This confirms Obj-Disco’s ability to detect any misaligned objectives signaling reward hacking.
>
> **Interpretable Alignment Dataset Selection**
>
> We applied Obj-Disco (DeBERTaV3 RM, GRPO, Llama-8B) across four alignment datasets and found distinct behavioral profiles:
>
> - HH-RLHF: *factual accuracy*, *reducing speculation*, *reducing inappropriate content* — safety-conscious helpfulness.
> - TLDR: *narrative coherence*, *concise summarization*, *conversational tone* — concise, natural delivery.
> - Skywork: *step-by-step reasoning*, *procedural clarity*, *answer completeness* — structured problem-solving.
> - Stanford Human Preferences: *domain-specific objectives* (e.g., ingredient recommendations, legal/ethical reasoning) — reflecting its broad, multi-domain nature.
>
> This shows Obj-Disco as a practical dataset selection tool: developers can compare behavioral profiles and choose datasets based on desired objectives (e.g., Skywork for reasoning, HH-RLHF for safety).
>
> ## Stress-Testing Obj-Disco (W1, Q3)
>
> **(More) K=5 Fine-Grained Objectives:**
>
> We tested with 5 ground-truth objectives and compared objectives.
>
> | Method | Precision | Recall | F1 |
> |---|---|---|---|
> | Obj-Disco | 100% | 93.3% | 96.3% |
> | Iter-Filter | 93.3% | 80.0% | 85.9% |
> | Zero-Shot | 100% | 93.3% | 96.3% |
>
> **"Needle-Like" Objectives:** We replaced one TLDR objective with extreme test cases. For "Do not use the letter 'e'," Obj-Disco does not detect this verbatim but consistently surfaces higher-level manifestations, e.g., "increase overall response brevity and terseness." For "State the main takeaway in the first sentence," Obj-Disco (k=10) discovered related objectives: "increase explicit articulation of user doubts and concerns" and "increase conversion of user posts into clear questions." In both cases, manual inspection of the optimized policy confirms the discovered objectives accurately reflect actual policy behavior. We note that optimizing such unusual objectives is itself a significant challenge for the underlying RL algorithm, meaning the policy may not fully realize the intended behavior; disentangling discovery from optimization limitations is outside our scope.
>
> **Competing Objectives:** Results in rebuttal to reviewer XcT5.
>
> **Directly Comparing Objectives:** Rebuttal to EesL.
>
> # Ablations (W2, Q4)
>
> **Ablating k parameter:** Results in rebuttal to reviewer vBsw.
>
> **Ablating trend predictability:** Results in rebuttal to reviewer XcT5.
>
> **Appendix A.5 already contains ablations over the composition function, number of parallel trajectories, number of candidate objectives, informative vs random samples, importance of model trajectory, and open-ended vs fixed space discovery**
>
> # Presentation / Figure 3 Update (W3, Q2)
>
> We will update our paper with the more thorough empirical results above, make our Intro more concise, and clearly state our assumptions. Updated Figure 3 is at this anonymized link: https://drive.google.com/file/d/1xG7K3RDRa92jBfcD90oF0e4RghJgoKqi/view?usp=sharing
>
> # Baselines (Q5)
>
> We confirm we are not aware of existing methods which can serve as direct baselines. Importantly, Obj-Disco discovers objectives as realized by the learned policy, not merely as encoded in the reward model (RM). Prior works analyzing the RM in isolation cannot determine whether incentives are actually exploited by the policy during training. Obj-Disco bridges this gap by analyzing the behavioral trajectory, which is especially valuable for detecting reward hacking, a phenomenon that emerges in policy behavior, not solely in reward specification.

---

> > ### Author Rebuttal · Reviewer_YTdq · 2026-04-03
> >
> > Please see my response in the "official comment" above

---

> > > ### Author Response · Authors · 2026-04-06
> > >
> > > Thank you very much. We aim to resolve your remaining concerns below. Added new experiments to surface novel insights:
> > > 1. Cross-reward-model comparison revealing novel and distinct behavioral profiles
> > > 2. Cross-policy-model comparison
> > >
> > > # Novel Insights
> > >
> > > Obj-Disco discovers objectives as realized by the learned policy, not merely as encoded in reward (initial rebuttal, EesL). Realized objectives are jointly shaped by reward, policy model/algorithm, and dataset, each with own inductive biases. How these interact to produce specific learned behaviors is poorly understood [1]: developers cannot predict which objectives a policy will internalize for a given config. Obj-Disco closes this gap by producing interpretable profiles for any combination, making most profiles a novel characterization.
> > >
> > > ## Experiment 1
> > > Obj-Disco (GRPO, HH-RLHF, Llama-8B) across different reward models.
> > >
> > > - DeBERTaV3. Objectives centered on content quality: "increase factual accuracy and correctness of information". Safety behavior manifests as passive content filtering ("reduce explicit and inappropriate content generation").
> > >
> > > - Skywork-3B. Active safety behavior, i.e. "avoid engaging with or clarifying harmful or sensitive queries", "enhance refusal and caution in response to unethical requests".
> > >
> > > - Skywork-1B. Different safety mechanism: passive hedging ("increase cautious ambiguity when addressing controversial topics") rather than 3B's active refusal.
> > >
> > > ## **Novel Finding**
> > > Not previously documented (to our knowledge):
> > > - In our setting, DeBERTaV3 prioritizes informativeness, Skywork-3B prioritizes structured safety, Skywork-1B prioritizes passive hedging
> > >
> > > ## Experiment 2
> > >
> > > Obj-Disco across 4 policy models of varying size and family (3 trials).
> > >
> > > - Llama-8B (same as DeBERTaV3 in Experiment 1).
> > >
> > > - Qwen-4B. More safety-oriented profile: "reduce inclusion of unsafe or unethical suggestions". Llama-8B did not surface ethics-related objectives.
> > >
> > > - Qwen-0.6B. Reveals behavior of a capacity-constrained model: surfacing "increase verbosity and repetitive content generation", suggesting struggles with repetition during alignment.
> > >
> > > - Qwen-1.7B. In middle but emphasizes bias mitigation: "reduce potentially offensive or biased language" emerges as highest coefficient across all models. Also uniquely surfaces "increase balanced presentation of nuanced information", suggesting preference for balanced discourse.
> > >
> > > ## **Novel Findings**
> > > 1. In our setting, safety-related objectives emerge with high weight in Qwen family but not in Llama, revealing family-specific biases
> > > 2. In our setting, Qwen 0.6B-4B scale affects objectives: 0.6B exhibits repetition-related objectives, while larger ones exhibit higher-level ones like vocabulary diversity or cultural sensitivity
> > >
> > > As far as we are aware, the above insights are novel.
> > >
> > > ## Misalignment Results
> > > Shows our method is **capable** of detecting the salient objectives according to reward (whereas baselines fail)
> > >
> > > # Stress Tests
> > >
> > > ## Method’s Intention
> > >
> > > Obj-Disco discovers objectives actually learned by policy, not theoretical reward intents. This is important:
> > > 1. Reward hacking emerges in policy behavior, not solely in reward, making policy-grounded discovery essential for detection (eg misalignment results)
> > > 2. Empirical gap between theoretical reward and actually learned objectives, an active research area [2]
> > >
> > > Needle-like setting: instilling "Do not use the letter 'e'" into policy is a challenging optimization problem [2] outside our scope. Manual inspection confirmed policy did not fully learn this behavior, so Obj-Disco correctly discovered the actually realized: "increase overall response brevity and terseness." This is optimization limitation, not discovery failure.
> > >
> > > ## K=5 Setting
> > > Truth: Thoroughness, Ethical, Clarity, Friendliness, Factual grounding
> > >
> > > Discovered:
> > > 1. Enhance factual accuracy (Factual)
> > > 2. Reduce redundant info (Clarity)
> > > 3. Minimize harmful actions (Ethical)
> > > 4. Enhance contextual relevance (Thoroughness)
> > > 5. Neutral stance on controversial topics (Friendliness)
> > >
> > > Objective (obj) space is exponentially large and we start with no predefined set. With k=5, the true objs are harder to specify independently, yet Obj-Disco recovers all 5 as disentangled axes. Baseline methods' objs often overlap. Combined with our strong quantitative results (Sec5), this shows robustness in challenging settings.
> > >
> > > # Discovered vs True Objectives
> > >
> > > HH-RLHF (True: Thoroughness, Ethical, Clarity)
> > > - Llama: "enhance contextual relevance in responses" (Thoroughness), "increase ethical response generation" (Ethical), "improve clarity and organization of information" (Clarity)
> > > - Qwen: "enhance context awareness in responses" (Thoroughness), "increase sensitivity to controversial topics" (Ethical), "enhance clarity and coherence in communication" (Clarity)
> > >
> > > [1] Singhal et al, A Long Way to Go: Investigating Length Correlations in RLHF, COLM 2024
> > >
> > > [2] Lu et al, Uncovering Cross-Objective Interference in Multi-Objective Alignment, 2026

---

### Decision · Program_Chairs · 2026-04-30

**Decision:**

Accept (regular)

**Comment:**

This paper studies reward models and seeks to identify interpretable objectives that explain reward model preferences. The authors demonstrate that the approach can be used to identify rewards that incentivize misaligned behaviors and that the objectives they identify are interpretable. Reviewers appreciated that this tackles an important problem and that it provides qualitatively sensible objectives. They raised concerns about the strength of the empirical results, the need for additional ablation studies, and concerns about the presentation of the underlying method. The authors engaged with the rebuttal, which reviewers found convincing and led to an increase in score. Based on the evaluations from reviewers, this paper makes an interesting, and novel contribution to a problem of interest. However, concerns remain about the level of evaluation and the ability of this approach to demonstrate novel and valuable insights into reward model behavior.